# Diagnostic Performance of the Acute Kidney Injury Baseline Creatinine Equations in Children and Adolescents with Type 1 Diabetes Mellitus Onset

**DOI:** 10.3390/diagnostics12102268

**Published:** 2022-09-20

**Authors:** Pier Luigi Palma, Stefano Guarino, Anna Di Sessa, Giulio Rivetti, Annalisa Barlabà, Federica Scaglione, Daniela Capalbo, Alfonso Papparella, Emanuele Miraglia del Giudice, Pierluigi Marzuillo

**Affiliations:** Department of Woman, Child and of General and Specialized Surgery, Università degli Studi della Campania “Luigi Vanvitelli”, Via Luigi De Crecchio 2, 80138 Naples, Italy

**Keywords:** acute kidney injury, type 1 diabetes mellitus, creatinine, estimation

## Abstract

Three new equations for calculating the estimated basal serum creatinine (ebSCr) in hospitalized children have been developed: the simplified acute kidney injury (AKI) baseline creatinine (ABC) equation which considered only age in the formula; the equation including age and minimum creatinine (Cr_min_) within the initial 72 h from hospitalization (ABC-cr); and the equation including Cr_min_ and height, weight, and age as squared values (ABC-advanced). We aimed to test the diagnostic performance of the ABC, ABC-cr and ABC-advanced equations in diagnosing AKI in 163 prospectively enrolled children with type 1 diabetes mellitus (T1DM) onset. We considered measured basal serum creatinine (mbSCr), the creatinine measured 14 days after T1DM onset. AKI was defined by the highest/basal serum creatine (HC/BC) ratio > 1.5. On the basis of the mbSCr, the AKI was diagnosed in 66/163 (40.5%) patients. This prevalence was lower than the prevalence of AKI diagnosed on the basis of ABC ebSCr (122/163 patients; 74.8%) (*p* < 0.001) and similar to the prevalence of AKI diagnosed on the basis of ABC-cr ebSCr (72/163 patients; 44.2%) (*p* = 0.5) and to the prevalence of AKI diagnosed on the basis of ABC-advanced ebSCr (69/163; 42.3%) (*p* = 0.73). AKI determined using ABC ebSCr, ABC-cr ebSCr and ABC-advanced ebSCr showed, respectively, 63.5% (kappa = 0.35; *p* < 0.001), 87.7% (kappa = 0.75; *p* < 0.001), and 87.1% (kappa = 0.74; *p* < 0.001) agreement with AKI determined using mbSCr. Using the HC/BC ratio calculated on the basis of mbSCr as gold standard, for Bland–Altman plots the HC/BC ratio calculated on the basis of ABC formula presented higher bias and wider limits of agreement compared with the HC/BC ratio calculated on the basis of ABC-cr and ABC-advanced formulas. In the receiver–operating characteristics (ROC) curve analysis the HC/BC ratio calculated on the basis of ABC ebSCr presented lower area under the ROC curve (AUROC) (AUROC = 0.89; 95%CI: 0.85–0.95; *p* < 0.001) compared with HC/BC ratio calculated on the basis of ABC-cr (AUROC = 0.94; 95%CI: 0.91–0.98; *p* < 0.001) or ABC-advanced ebSCr (AUROC = 0.914; 95%CI: 0.91–0.97; *p* < 0.001). In both Bland–Altman plots and ROC curve analysis, the ABC-cr and ABC-advanced formulas performed similarly. In conclusion, the ABC-cr and ABC-advanced formulas present very good diagnostic performance toward AKI identification in a population of children with T1DM onset.

## 1. Introduction

Acute kidney injury (AKI) could be a complication of several common pediatric conditions [1,2,3] but it is often under-recognized in hospitalized children [4]. AKI spontaneously reverts most of the time [1,2,3,5] without needing hemodialysis. The spontaneous AKI improvement gives reassurance about a possible AKI under-recognition in the short term. In the long-term, however, the effects of AKI on kidney health are alarming. In fact, having presented a mild AKI doubles the risk of chronic kidney disease (CKD) later in life [6]. This risk further increases with the increase in the AKI severity [6]. A specific nephrological follow-up after AKI improves outcomes [7], and this is why it is mandatory not to miss AKI episodes in children, in order to plan an accurate post-AKI follow-up.

Children with type 1 diabetes mellitus (T1DM) have an increased risk of developing kidney involvement, both in the acute setting presenting with AKI and in the chronic setting presenting with diabetic kidney disease (DKD) [8]. In children with T1DM onset, AKI may occur in about 40% of the cases, with a prevalence increasing up to 65% in the case of diabetic ketoacidosis (DKA) and up to 81% in the case of recurrent DKA [8]. Therefore, it is pivotal to correctly diagnose AKI in these patients to plan an accurate follow-up and to start timely treatment with renin-aldosterone system inhibitors in the case of proteinuria or hypertension [8].

According to the Kidney Disease/Improving Global Outcomes (KDIGO) guidelines, in order to diagnose AKI, it is important to know the measured basal serum creatinine (mbSCr). The AKI under-recognition in children could be due to the lack of a known mbSCr in most of children. Absence of AKI, in fact, is defined by creatinine values <1.5 times the mbSCr, stage 1 AKI by creatinine values from 1.5 to <2, stage 2 from 2 to <3, and stage 3 if creatinine values are ≥3 times the bSCr [9].

To allow an AKI diagnosis when the mbSCr is unknown, an estimated bSCr (ebSCr) can be back-calculated [10]. In a previous study we validated the use of ebSCr back-calculated on the basis of patients’ height to diagnose AKI in a population of children with T1DM onset [11].

Recently, Braun et al. developed and validated new equations, the AKI baseline creatinine (ABC) equations, for calculating the ebSCr in hospitalized children and adolescents without pre-existing kidney disease [12]. These authors mainly proposed three equations: the simplified equation (ABC) which considered only age in the formula, the equation including age and minimum creatinine (Cr_min_) within the initial 72 h from hospitalization (ABC-cr), and the equation including Cr_min_ within 72 h of admission and height, weight, and age as squared values (ABC-advanced) [12]. The ABC-cr and ABC-advanced more accurately estimated ebSCr by ≥25% compared with previous methods in a general pediatric population hospitalized for any reason [10].

We aimed to test the diagnostic performance of the ABC, ABC-cr and ABC-advanced equations in diagnosing AKI in children with T1DM onset to validate the use of these formulas in a population of children in which the risk of CKD is particularly high and in which a proper and accurate AKI diagnosis is important to timely diagnose and treat complications during follow-up. For this purpose, we used the DiAKIdney (T1DM and AKI) cohort [5], a prospectively enrolled cohort of children during T1DM onset and with availability of mbSCr admitted in a non-PICU ward experienced in the T1DM management.

## 2. Methods

The DiAKIdney cohort represents a cohort of 185 prospectively enrolled children with T1DM onset [5]. These patients were evaluated both at T1DM onset and 14 days later when they fully recovered from the acute phase.

Inclusion criteria of the DiAKIdney cohort were: (i) onset of T1DM; (ii) age < 18 years; and (iii) not on any medication apart from intravenous 0.9% NaCl infusion. Exclusion criteria were: (i) not returning for the scheduled follow-up; and (ii) anomalies of the kidney and urinary tract.

For the analyses of this manuscript we excluded 22 patients with age < 2 years, in order to only include patients with mature renal function [13]. Therefore, 163 patients with positive glutamic acid decarboxylase and/or islet antigen 2 and/or insulin and/or zinc transporter 8 antibodies at the time of diagnosis were included in this study.

DKA was defined by blood glucose level ≥ 200 mg/dL, pH ≤ 7.3 or bicarbonates ≤ 15 mEq/L, and elevation of serum ketones [14].

We measured the serum levels of creatinine at admission, every 24 h for 3 days, and then after 5 and 14 days. In our laboratories the creatinine has been and is usually estimated with the Jaffe method (methodology: Alkaline Picrate, Abbott catalog no. 7D64-20) by using the Architect c16000 automated analyzer (Abbott Diagnostics Inc, Park City, IL, USA) [5,15,16].

### 2.1. AKI Definition

In accordance with the KDIGO criteria [9], AKI was defined by the ratio of highest creatinine/basal creatinine values (HC/BC) > 1.5. For mbSCr, we used the creatinine measurement obtained after 14 days following the T1DM onset when the DiAKIdney patients fully recovered kidney function and presented good T1DM control [5].

### 2.2. ebSCr Calculation

To calculate the ebSCr, we used the new formulas to estimate basal creatinine, as proposed by Braun et al. [12]. More in detail, using the ABC equation: ebSCr = 0.264 × e^0.056xAge^; using the ABC-cr equation: ebSCr = 0.578 × [Cr_min_]^0.585^ × e^0.0259×Age^; and using ABC-advanced: ebSCr = 0.606 × [Cr_min_]^0.5302^ × e [–0.0029 × Height + 0.000021 × Height^2^ + 0.000732 × Weight − 0.00000387 × Weight^2^ + 0.0215 × Age − 0.000455 × Age^2^].

### 2.3. Statistical Analysis

*p* values < 0.05 were considered statistically significant. While data are shown as means ± standard deviation score (SDS), differences for continuous variables were analyzed with independent sample t test, if normally distributed, and with the Mann–Whitney test in the case of non-normally distributed variables. By linear regression analyses we assessed the relationship between HC/BC calculated on the basis of mbSCr, and HC/BC calculated on the basis of ebSCr (both using ABC and ABC-cr or ABC-advanced formulas). Moreover, correlations between HC/BC calculated on the basis of mbSCr and HC/BC calculated on the basis of ebSCr (both using ABC and ABC-cr or ABC-advanced formulas) were assessed by Pearson test.

To evaluate the percentage of agreement in AKI classification (yes/no) by ebSCr and mbSCr, kappa statistics was used.

Bland–Altman plots were used to show the agreement between the HC/BC ratio calculated by using mbSCr or ebSCr as the denominator.

The area under receiver–operating characteristic (AUROC) of HC/BC ratio calculated on the basis of ebSCr (both using ABC and ABC-cr or ABC-advanced formulas) toward presence of AKI, was evaluated. The AUROC were compared using the Hanley and McNeil test [17]. SPSS 25 software and GraphPad Prism 8.0 for Windows were used for all the statistical analyses.

## 3. Results

In this study, 163 Caucasian children with T1DM onset (91 of female gender) and with a mean age of 9.9 years (3.4 SDS) were included. Out of 163 patients, 83 (50.9%) presented with DKA.

On the basis of the mbSCr (gold standard), the AKI was diagnosed in 66 out of 163 (40.5%) patients. This prevalence was lower than the prevalence of AKI diagnosed on the basis of ABC ebSCr (122/163 patients; 74.8%) (*p* < 0.001), while it was similar to the prevalence of AKI diagnosed on the basis of ABC-cr ebSCr (72/163 patients; 44.2%) (*p* = 0.5), and to the prevalence of AKI diagnosed on the basis of ABC-advanced ebSCr (69/163; 42.3%) (*p* = 0.73).

### Performance of the ebSCr Calculated on the Basis of the Examined Formulas Compared with mbSCr in Diagnosing AKI

The HC/BC ratio calculated on the basis of the mbSCr (1.43 ± 0.38) was lower than both ABC ebSCr (1.86 ± 0.53; *p* < 0.001) (Figure 1) and ABC-cr ebSCr (1.55 ± 0.39; *p* = 0.005) (Figure 2), or ABC-advanced ebSCr (1.53 ± 0.39; *p* = 0.02) (Figure 3).

The HC/BC ratio calculated on the basis of ABC ebSCr, ABC-cr ebSCr, and ABC-advanced ebSCr showed good correlation with HC/BC ratio calculated on the basis of mbSCr both at Pearson test (r = 0.86 and *p* < 0.001, r = 0.91 and *p* < 0.001, r = 0.90 and *p* < 0.001, respectively) and regression analyses (Figure 4, Figure 5, and Figure 6, respectively).

The Bland–Altman graphs show the values of both the superior and inferior limits, and the average of the differences of the HC/BC ratio calculated on the basis of the mbSCr from the ebSCr measured with the different formulas. Compared with the HC/BC ratio calculated on the basis of mbSCr, for the HC/BC ratio calculated on the basis of ABC ebSCr, the bias was −0.42 (95% limits of agreement from −0.98 to 0.13) (Figure 7); for the HC/BC ratio calculated on the basis of ABC-Cr ebSCr, the bias was −0.12 (95% limits of agreement from −0.45 to 0.20) (Figure 8); while for the HC/BC ratio calculated on the basis of the ABC-advanced ebSCr, the bias was −0.10 (95% limits of agreement from −0.44 to 0.24) (Figure 9).

AKI determined using ABC ebSCr, ABC-cr ebSCr and ABC-advanced ebSCr showed, respectively, 63.5% (kappa = 0.35; *p* < 0.001), 87.7% (kappa = 0.75; *p* < 0.001) and 87.1% (kappa = 0.74; *p* < 0.001) agreement with AKI determined using mbSCr.

Finally, we evaluated the AUROC of the HC/BC ratio calculated on the basis of ABC ebSCr (AUROC = 0.89, 95%CI: 0.85–0.95; *p* < 0.001), ABC-cr ebSCr (AUROC = 0.94, 95%CI: 0.91–0.98; *p* < 0.001) and ABC-advanced ebSCr (AUROC = 0.914, 95%CI: 0.91–0.97; *p* < 0.001) toward the AKI diagnosis on the basis HC/BC ratio calculated using the mbSCr (Figure 10).

The AUROC of the HC/BC ratio calculated on the basis of ABC ebSCr was lower compared with the AUROC of the HC/BC ratio calculated on the basis of ABC-cr ebSCr (*p* < 0.001) and ABC-advanced ebSCr (*p* < 0.001). On the other hand, the AUROCs of the HC/BC ratio calculated on the basis of ABC-cr ebSCr and ABC-advanced ebSCr performed similarly (*p* = 0.71).

## 4. Discussion

In patients with T1DM, a particular care of kidney health should be taken. In addition to the risk related to a poor metabolic control of the T1DM, another part of the risk is played already at T1DM onset [8]. Up to 65% of the patients at T1DM onset develop AKI, which, in turn, is associated with CKD later in life [8]. The risk of CKD is higher in the case of more severe AKI or in the case of recurrent DKA associated with recurrent AKI episodes [8]. In addition, AKI itself is associated with hypertension, which, in turn, predisposes the evolution toward DKD and CKD [8]. The current guidelines suggest starting screening for DKD at 11 years of age with 2–5 years T1DM duration [18]. However, considering the impact of AKI on the development of microalbuminuria and then DKD [19], it has been suggested to anticipate the screening for DKD with a closer follow-up for patients who developed AKI at T1DM onset, and especially in case of recurrent DKA [8]. A proper post-AKI nephrological follow-up, in fact, improves outcomes [7].

For these reasons, an accurate AKI diagnosis is pivotal. Recently—in a population of children with T1DM onset—we demonstrated that by back-calculating creatinine on the basis of height by a previously validated method [10], we can obtain a good agreement (kappa = 0.66) with mbSCr. In the current study, we also validated the utilization of the recent ABC formulas [12]. These formulas were developed in a retrospectively enrolled population of patients aged 0–25 years and hospitalized for any reason. Interestingly, we demonstrated that these formulas also run well in calculating ebSCr in a selected and prospectively enrolled population of children with T1DM onset. Not all three formulas, however, performed similarly. The ABC formula presented a poorer diagnostic performance (kappa = 0.35) toward AKI compared with ABC-cr (kappa = 0.75) and ABC-advanced (kappa = 0.74).

Similarly, at Bland–Altman plots, the HC/BC ratio calculated on the basis of the ABC formula presented higher bias and wider limits of agreement compared with the HC/BC calculated on the basis of ABC-cr and on the basis of ABC-advanced formulas (Figure 7, Figure 8 and Figure 9), and in ROC curve analysis the HC/BC ratio calculated on the basis of ABC ebSCr presented lower AUROC compared with the HC/BC ratio calculated on the basis of ABC-cr or ABC-advanced ebSCr (Figure 10). On the other hand, in both Bland–Altman plots (Figure 7, Figure 8 and Figure 9) and ROC curve analysis (Figure 10), the HC/BC ratio calculated on the basis of ABC-cr and ABC-advanced formulas performed similarly.

Therefore, in our opinion, to diagnose AKI in children with T1DM onset and unknown mbSCr, the ABC-cr and ABC-advanced formulas should be preferred [12]. Considering that the ABC-cr is significantly easier to run on pocket calculators compared with the ABC-advanced formula, it is likely that the ABC-cr formula will find more extensive application in clinical practice. In any case, we developed a calculator on a excel file (Appendix A) [12], to assist clinicians in their daily clinical practice.

Limitation of our study is represented by the utilization of the Jaffe method to measure creatinine, in spite of IDMS-traceable methods in our cohort. However, the percentage of centers using Jaffe methods is not negligible. In fact, in 2019, 24% of laboratories surveyed by the College of American Pathologists were using the Jaffe method to measure serum creatinine [20]. On the other hand, this limitation could be also an added value of the study because we validated these equations developed on creatinine measured by enzymatic assays, with creatinine measured by the Jaffe method.

In conclusion, the ABC-cr and ABC-advanced formulas proposed by Braun et al. [12] present a very good diagnostic performance toward AKI identification in a population of children with T1DM onset. Considering that the ABC-cr formula is simpler to calculate compared with the ABC-advanced formula, the ABC-cr formula could be preferred in the daily clinical practice.

## Figures and Tables

**Figure 1 diagnostics-12-02268-f001:**
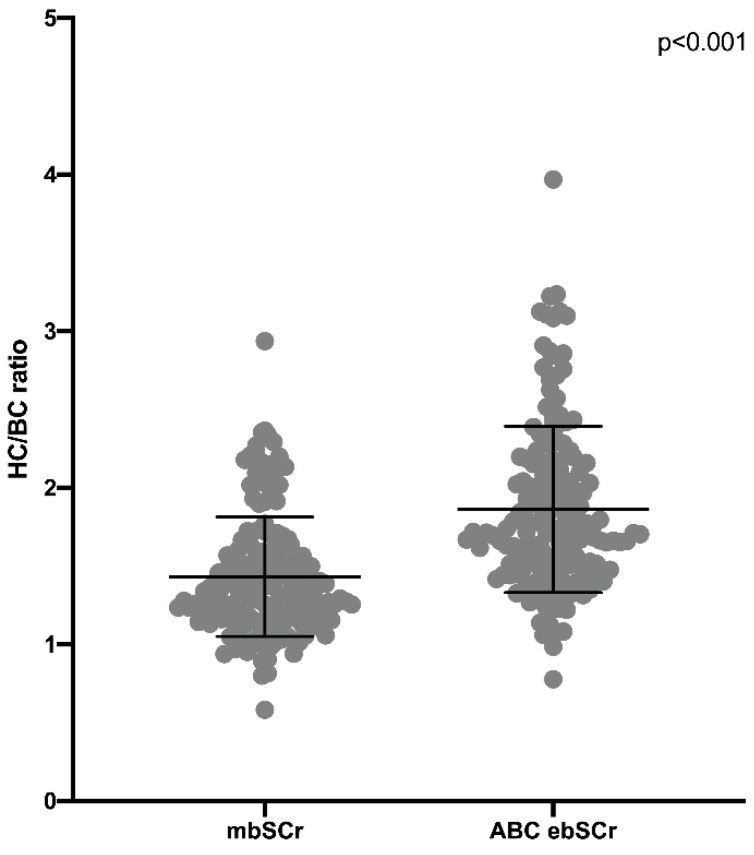
HC/BC ratio calculated on the basis of the mbSCr compared with HC/BC ratio calculated on the basis of ABC ebSCr.

**Figure 2 diagnostics-12-02268-f002:**
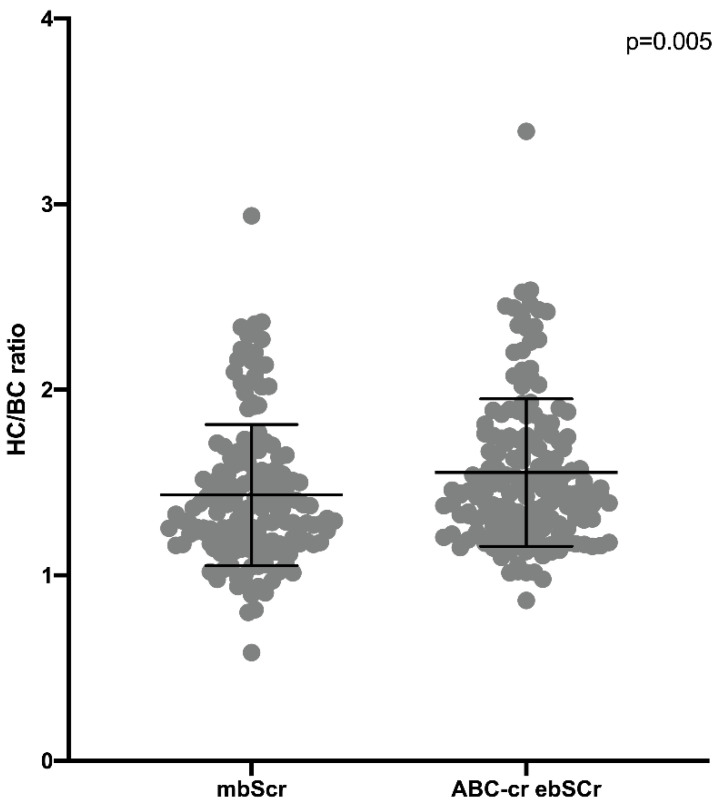
HC/BC ratio calculated on the basis of the mbSCr compared with HC/BC ratio calculated on the basis of ABC-cr ebSCr.

**Figure 3 diagnostics-12-02268-f003:**
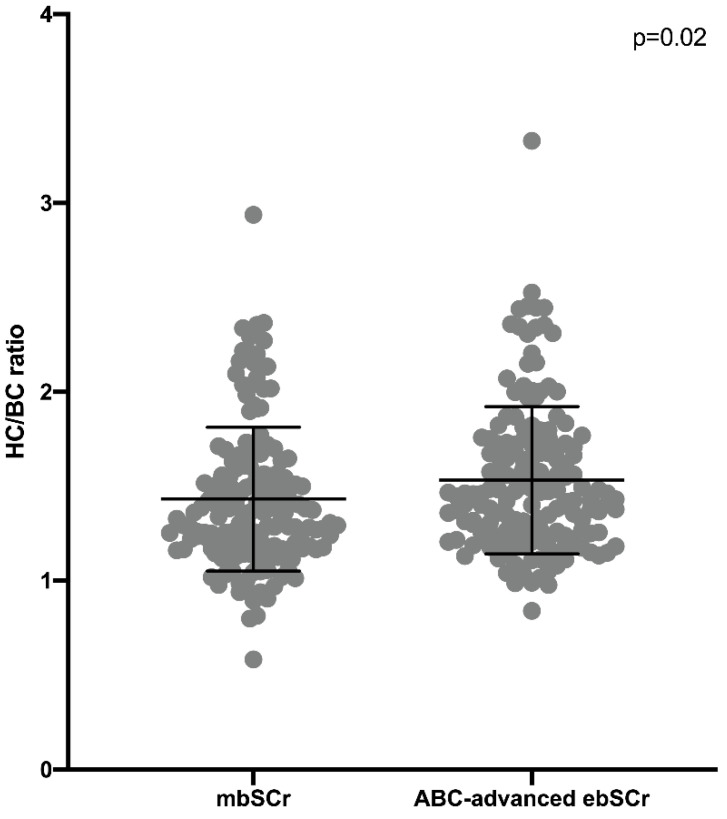
HC/BC ratio calculated on the basis of the mbSCr compared with HC/BC ratio calculated on the basis of ABC-advanced ebSCr.

**Figure 4 diagnostics-12-02268-f004:**
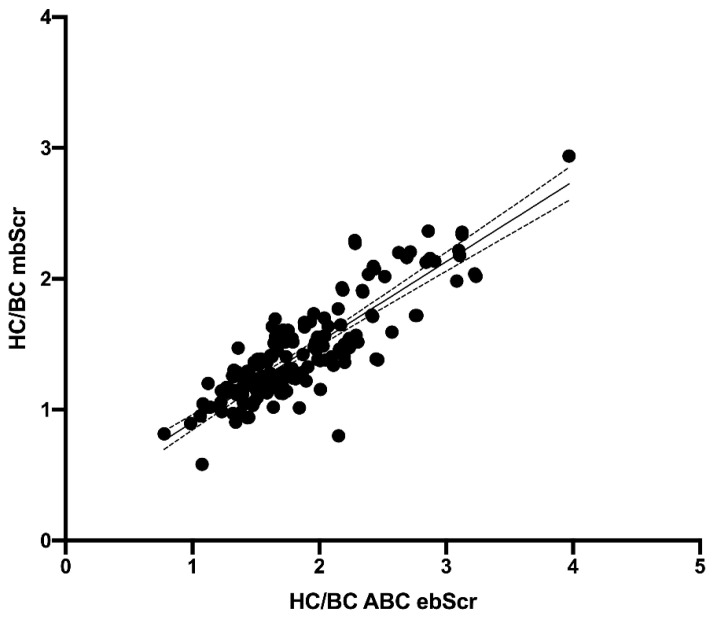
Regression analysis describing the relationship between HC/BC ratio calculated on the basis of mbSCr and ABC ebSCr. Model r2 = 73.2 percent; *p* < 0.001; correlation coefficient = 0.85. The regression is described by the equation Y = 0.6135*X + 0.2902.

**Figure 5 diagnostics-12-02268-f005:**
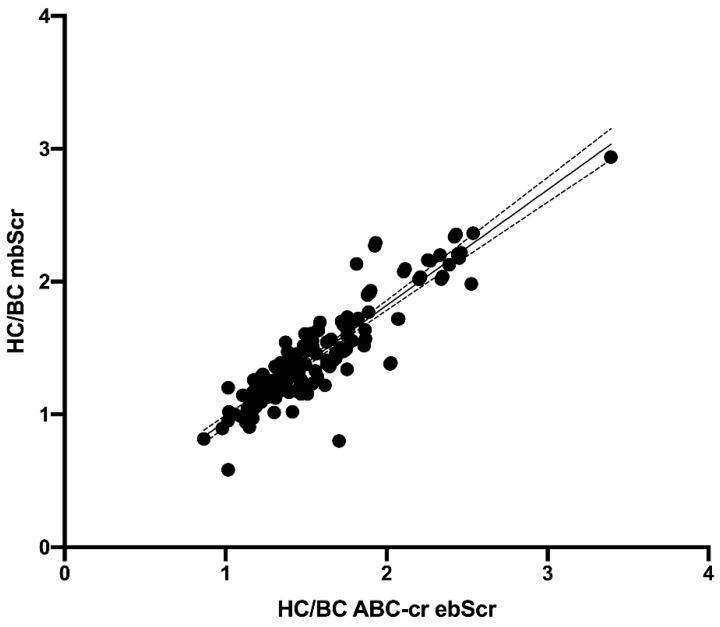
Regression analysis describing the relationship between HC/BC ratio calculated on the basis of mbSCr and ABC-cr ebSCr. Model r2 = 82.7 percent; *p* < 0.001; correlation coefficient = 0.91. The regression is described by the equation Y = 0.8709*X + 0.07908.

**Figure 6 diagnostics-12-02268-f006:**
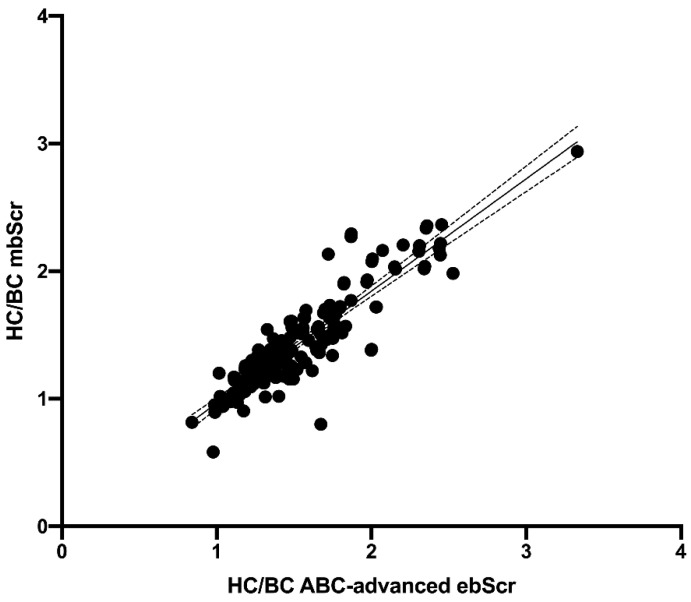
Regression analysis describing the relationship between HC/BC ratio calculated on the basis of mbSCr and ABC-advanced ebSCr. Model r2 = 81.2 percent; *p* < 0.001; correlation coefficient = 0.90. The regression is described by the equation Y = 0.8800*X + 0.08352.

**Figure 7 diagnostics-12-02268-f007:**
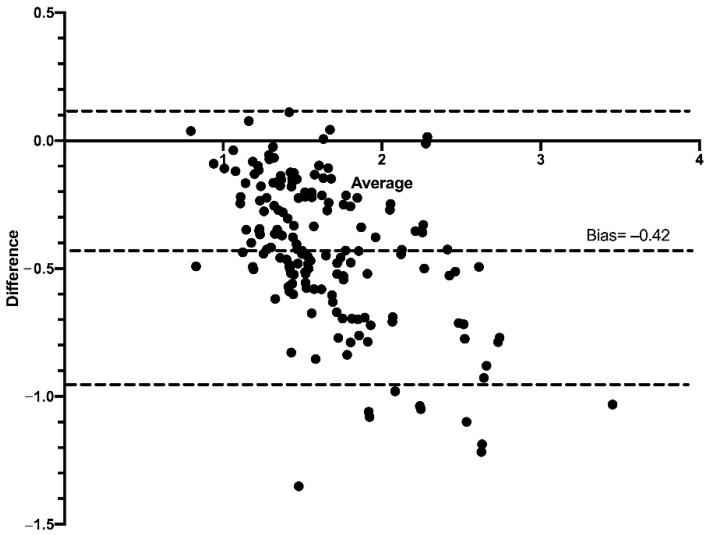
Bland–Altman plots comparing the HC/BC calculated on the basis of the mbSCr and on the basis of ABC ebSCr. Dashed lines represent the limits of agreement and mean difference (bias) in estimations.

**Figure 8 diagnostics-12-02268-f008:**
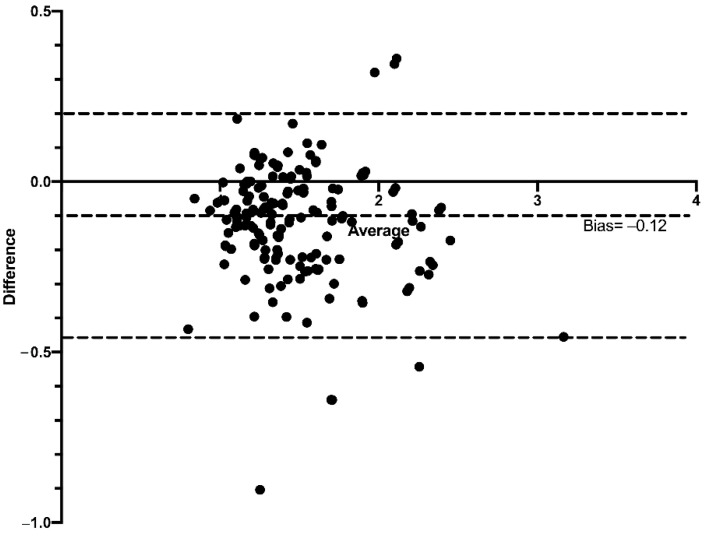
Bland–Altman plots comparing the HC/BC calculated on the basis of the mbSCr and on the basis of ABC-cr ebSCr. Dashed lines represent the limits of agreement and mean difference (bias) in estimations.

**Figure 9 diagnostics-12-02268-f009:**
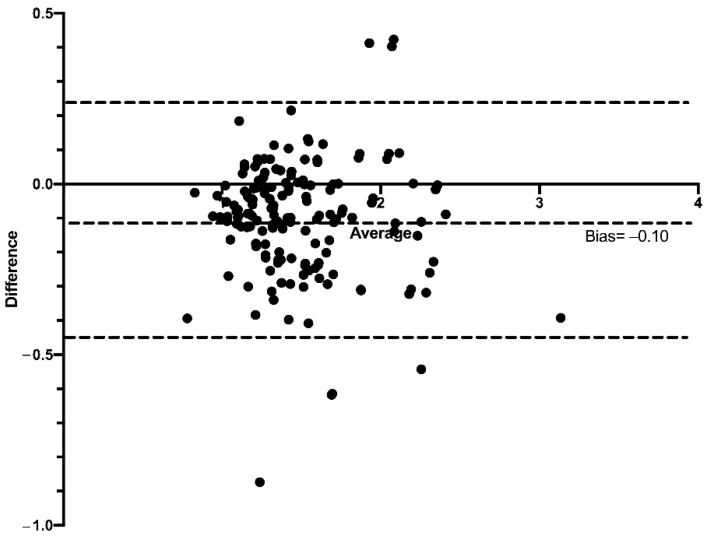
Bland–Altman plots comparing the HC/BC calculated on the basis of the mbSCr and on the basis of ABC-advanced ebSCr. Dashed lines represent the limits of agreement and mean difference (bias) in estimations.

**Figure 10 diagnostics-12-02268-f010:**
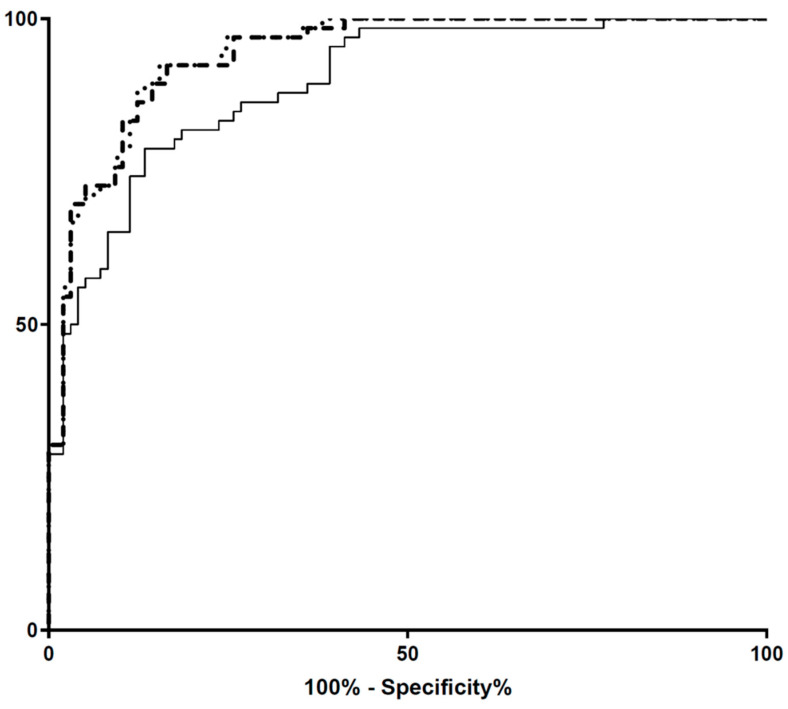
AUROC of the HC/BC ratio calculated on the basis of ABC ebSCr, ABC-cr ebSCr and ABC-advanced ebSCr toward the AKI diagnosis on the basis of the mbSCr. Dotted line: ROC curve of HC/BC ratio calculated on the basis of ABC-cr ebSCr. Dashed line: ROC curve of HC/BC ratio calculated on the basis of ABC-advanced ebSCr. Continuous line: ROC curve of HC/BC ratio calculated on the basis of ABC ebSCr.

## Data Availability

Data supporting reported results can be obtained on request.

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
