# Peer review of "Diagnostic Performance of the Acute Kidney Injury Baseline Creatinine Equations in Children and Adolescents with Type 1 Diabetes Mellitus Onset"

_diagnostics, 2022, doi:10.3390/diagnostics12102268_

Round 1
Reviewer 1 Report
The manuscript entitled “Diagnostic performance of the acute kidney injury baseline creatinine equations in children and adolescents with type 1 diabetes mellitus onset” evaluated the diagnostic performance of acute kidney injury baseline creatinine (ABC-cr) equations with T1DM children and adolescents. The authors recruited 163 and found that ABC-cr and ABC-advanced formulas present a very good diagnostic performance toward AKI identification in a population of children with T1DM onset. The manuscript has merit for publication. However, some modifications should be performed.
1. The patients' sociodemographic and clinicopathological were not mentioned.
2. The authors mentioned that “creatinine has been and is usually dosed with Jaffe method”. Jaffe method was usually used for the estimation of creatinine, not for dosing purposes.
3. Why did the authors recruit a higher number of female children than male children? Is the AKI more prevalent in females than males?
4. Many grammatical errors were detected throughout the manuscript that should be corrected.
Author Response
We thank the Editor and Reviewers for their time reviewing our manuscript and their helpful comments.
Reviewer 1
The manuscript entitled “Diagnostic performance of the acute kidney injury baseline creatinine equations in children and adolescents with type 1 diabetes mellitus onset” evaluated the diagnostic performance of acute kidney injury baseline creatinine (ABC-cr) equations with T1DM children and adolescents. The authors recruited 163 and found that ABC-cr and ABC-advanced formulas present a very good diagnostic performance toward AKI identification in a population of children with T1DM onset. The manuscript has merit for publication. However, some modifications should be performed.
Answer: thank you.
- The patients' sociodemographic and clinicopathological were not mentioned.
Answer: we added information about ethnicity, age, gender and prevalence of diabetic ketoacidosis in the new version of the manuscript. Please see lines 145-147 of the new version of the manuscript. We also added information about inclusion/exclusion criteria and setting of enrollment (please see lines 94-104 of the new version of the manuscript).
- The authors mentioned that “creatinine has been and is usually dosed with Jaffe method”. Jaffe method was usually used for the estimation of creatinine, not for dosing purposes.
Answer: we modified the text accordingly. Please see line 111 of the new version of the manuscript.
- Why did the authors recruit a higher number of female children than male children? Is the AKI more prevalent in females than males?
Answer: The number of females and males is fortuitous. These patients have been consecutively enrolled according to inclusion and exclusion criteria (please see lines 94-104 of the new version of the manuscript. There was no difference in the prevalence of AKI comparing female (36.6%) Vs male (45.8%) patients with p=0.22. We decided to not add this information in the new version of the manuscript because it is outside from the aims of this manuscript.
- Many grammatical errors were detected throughout the manuscript that should be corrected.
Answer: we carefully revised the written English of this manuscript.
Reviewer 2 Report
AKI diagnosis is highly important to be established on time (not only in children), therefore it is mandatory to identify the best diagnostic tools that could be easily used in daily practice. Furthermore, as the assessment of children with renal impairment could be sometimes difficult to be performed, it is necessary to determine the most suitable modality in determining AKI on time. In my opinion, your study provides useful information, and my only remark is related to text editing / grammar - please read again the text between lines 91-93, especially the end of the phrase (“age < 2 years to exclude patients with a non-mature renal function”).
Author Response
We thank the Editor and Reviewers for their time reviewing our manuscript and their helpful comments.
Reviewer 2
AKI diagnosis is highly important to be established on time (not only in children), therefore it is mandatory to identify the best diagnostic tools that could be easily used in daily practice. Furthermore, as the assessment of children with renal impairment could be sometimes difficult to be performed, it is necessary to determine the most suitable modality in determining AKI on time. In my opinion, your study provides useful information, and my only remark is related to text editing / grammar - please read again the text between lines 91-93, especially the end of the phrase (“age < 2 years to exclude patients with a non-mature renal function”).
Answer: we largely revised this sentence also accordingly to the comment 1 of the Reviewer 1 (please see lines 94-104 of the new version of the manuscript). We also edited the written English of all the manuscript.